# Causal relationships of Helicobacter pylori and related gastrointestinal diseases on Type 2 diabetes: Univariable and Multivariable Mendelian randomization

**Mei Sun**[1,2‡], **Zhe Zhang**[3,4‡], **Jingjing Zhang**[5‡], **Juewei Zhang**[6‡], **Zhuqiang Jia**[7,8], **Lin Zhao**[9], **Xin Han**[8,10], **Xiaohong Sun**[11], **Junwei Zong**[3]*, **Ying Zhu**[1]*, **Shouyu Wang**[3]*

1 Department of Infectious Diseases, The First Affiliated Hospital of Dalian Medical University, Dalian, China, 2 Department of gastroenterology, Dalian Municipal Central Hospital, Dalian, China, 3 Department of Orthopaedic Surgery, The First Affiliated Hospital of Dalian Medical University, Dalian, China, 4 College of Integrative Medicine, Dalian Medical University, Dalian, China, 5 Department of Gastroenterology, The Second Affiliated Hospital of Dalian Medical University, Dalian, China, 6 Health Inspection and Quarantine, College of Medical Laboratory, Dalian Medical University, Dalian, China, 7 The First Affiliated Hospital of Dalian Medical University, Dalian, China, 8 Naqu People's Hospital, Tibet, China, 9 Department of Quality Management, Dalian Municipal Central Hospital, Dalian, China, 10 Department of Orthopaedic Surgery, The Second Affiliated Hospital of Dalian Medical University, Dalian, China, 11 Department of Nursing, The First Affiliated Hospital of Dalian Medical University, Dalian, China

‡ MS, ZZ, JZ, and JZ have contributed equally to this work and share first authorship.
* aweizone@163.com (JZ); zhuyingsh52@126.com (YZ); wangshouyu666@126.com (SW)

**Data Availability Statement:** The minimal data for this study are publicly accessible from the GWAS

## Abstract

### Background

Previous observational studies have demonstrated a connection between the risk of Type 2 diabetes mellitus (T2DM) and gastrointestinal problems brought on by Helicobacter pylori (H. pylori) infection. However, little is understood about how these factors impact on T2DM.

### Method

This study used data from the GWAS database on H. pylori antibodies, gastroduodenal ulcers, chronic gastritis, gastric cancer, T2DM and information on potential mediators: obesity, glycosylated hemoglobin (HbA1c) and blood glucose levels. Using univariate Mendelian randomization (MR) and multivariate MR (MVMR) analyses to evaluate the relationship between H. pylori and associated gastrointestinal diseases with the risk of developing of T2DM and explore the presence of mediators to ascertain the probable mechanisms.

### Results

Genetic evidence suggests that H. pylori IgG antibody (P = 0.006, b = 0.0945, OR = 1.0995, 95% CI = 1.023–1.176), H. pylori GroEL antibody (P = 0.028, OR = 1.033, 95% CI = 1.004–1.064), gastroduodenal ulcers (P = 0.019, OR = 1.036, 95% CI = 1.006–1.068) and chronic gastritis (P = 0.005, OR = 1.042, 95% CI = 1.012–1.074) are all linked to an increased risk of T2DM, additionally, H. pylori IgG antibody is associated with obesity (P = 0.034, OR = 1.03,

database (https://gwas.mrcieu.ac.uk). To access the minimal data, please visit the following URLs: Helicobacter pylori CagA antibody levels: https://gwas.mrcieu.ac.uk/datasets/ebi-a-GCST90006911/; Helicobacter pylori Catalase antibody levels: https://gwas.mrcieu.ac.uk/datasets/ebi-a-GCST90006912/; Helicobacter pylori GroEL antibody levels: https://gwas.mrcieu.ac.uk/datasets/ebi-a-GCST90006913/; Helicobacter pylori OMP antibody levels: https://gwas.mrcieu.ac.uk/datasets/ebi-a-GCST90006914/; Helicobacter pylori UREA antibody levels: https://gwas.mrcieu.ac.uk/datasets/ebi-a-GCST90006915/; Helicobacter pylori VacA antibody levels: https://gwas.mrcieu.ac.uk/datasets/ebi-a-GCST90006916/; Gastroduodenal ulcer: https://gwas.mrcieu.ac.uk/datasets/finn-b-K11_GASTRODUOULC/; Chronic gastritis: https://gwas.mrcieu.ac.uk/datasets/finn-b-K11_CHRONGASTR/; Malignant neoplasm of stomach: https://gwas.mrcieu.ac.uk/datasets/finn-b-CD2_BENIGN_STOMACH/; Obesity: https://gwas.mrcieu.ac.uk/datasets/ieu-a-92/; HbA1c: https://gwas.mrcieu.ac.uk/datasets/ukb-d-30750_raw/; Blood glucose levels: https://gwas.mrcieu.ac.uk/datasets/ebi-a-GCST90025986/; Type 2 diabetes: https://gwas.mrcieu.ac.uk/datasets/finn-b-E4_DM2NASCOMP/.

**Funding:** This study was supported by the National Natural Science Foundation of China (82074426, 82104864, 82204822), Natural Science Foundation of Liaoning Province (2021-BS-215, 2022-MS-25, 2023-MS-13), Liaoning Revitalization Talents Program (XLYC1802014), Key Research and Development Program of Liaoning Province (2017226015), Natural Science Foundation of Tibet Autonomous Region (XZ202301ZR0030G, XZ2023ZR-ZY82(Z)). There was no additional external funding received for this study.

**Competing interests:** The authors have declared that no competing interests exist.

95% CI = 1.002–1.055). The results of MVMR showed that the pathogenic relationship between H. pylori GroEL antibody and gastroduodenal ulcer in T2DM is mediated by blood glucose level and obesity, respectively.

## Conclusion

Our study found that H. pylori IgG antibody, H. pylori GroEL antibody, gastroduodenal ulcer and chronic gastritis are all related to t T2DM, and blood glucose level and obesity mediate the development of H. pylori GroEL antibody and gastroduodenal ulcer on T2DM, respectively. These findings may inform new prevention and intervention strategies for T2DM.

## Introduction

Type 2 diabetes mellitus (T2DM) is a disease defined by high blood sugar levels and a relative shortage of insulin, as well as insulin resistance and other chronic metabolic illnesses caused predominantly by diabetes, accounting for more than 90% of diabetic patients [1]. Patients with T2DM have a more than 50% chance of developing complications [2], and some studies shows that at least 80% of diabetic patients will die from cardiac complications [3, 4]. Even though T2DM is better understood and treated than it formerly was, the disease's incidence and prevalence are still rising internationally. Researchers from Sweden made the initial discovery of H. pylori in a culture of human stomach mucosa [5]. Around 60% of people on the planet have H. pylori infections. Cell degeneration, necrosis, and inflammatory cell infiltration can be caused by the inflammation and immunological response brought on by H. pylori infection, and specific antibodies can be found in the serum. TLR1 is primarily localized on the cellular surface, exhibiting heightened expression on immune cells, including macrophages and dendritic cells. It often forms a heterodimer with TLR2, collaboratively engaging in immune responses. The FCGR2A gene, which is classified as a low-affinity receptor for immunoglobulin G (IgG). This receptor is predominantly expressed across various cells within the immune system, encompassing B cells, NK cells, macrophages, and dendritic cells. Both genes bear a significant association with Helicobacter pylori infection [6]. H. pylori is closely related to gastritis, peptic ulcer, gastric cancer and other diseases [7], and H. pylori has been included in the list of a class of carcinogens by the World Health Organization.

T2DM has been recognized as one of the probable illnesses of H. pylori infection in a broad range of observational studies, with 61.5% of H. pylori positive patients having one or more chronic diabetic sequelae [8]. In T2DM patients with concurrent H. pylori infection, H. pylori antibodies could be detected, and 75% of patients experienced gastrointestinal symptoms [9]. H. pylori infection appears to be linked to chronic inflammation, impaired insulin secretion, and higher mean glycosylated hemoglobin levels, all of which increase the risk of T2DM [10, 11]. Additionally, it has been noted that among T2DM patients, obese people have a much greater frequency of H. pylori infection than non-obese patients. However, the correlation between H. pylori infection and the risk of T2DM is still debatable. According to some studies [11–13], there is no distinction between diabetic and nondiabetic people in terms of the prevalence of H. pylori infection. This finding may be due to sources of potential bias in observational studies, such as reverse causal consistency and confounding. Therefore, stronger proof is required to show a connection between H. pylori and T2DM.

Typical observational studies cannot demonstrate causation or rule out the impact of confounding factors; they can only show whether there is a correlation between two variables. An

epidemiological analytic technique called Mendelian randomization (MR) makes it easier to deduce causes. MR designs use single nucleotide polymorphisms (SNPs), which have a random distribution without being impacted by outside influences and other confounders, as instrumental variables (IVs) for target exposures [14]. As a result, complex disease causal linkages can be rigorously explained using MR designs. MR has been applied more frequently as a result of the expansion of genome-wide association studies (GWAS) and the accessibility of massive amounts GWAS data. In our study, we used seven H. pylori antibodies, as well as chronic gastritis, gastroduodenal ulcer, and gastric cancer, which are closely associated with H. pylori, as exposures, and T2DM as an outcome. Then, we chose obesity, glycosylated hemoglobin (HbA1c), and blood glucose levels as potential mediators to clarify the association of H. pylori and its associated digestive diseases on T2DM and see whether there is a mediating effect, in the hopes of elucidating the causality and offering beneficial treatment and diagnosis recommendations.

## Materials and methods

### 1. Mendelian randomized design

The Structure of the current MR study is shown in Fig 1, and to find the connections between H. pylori and related gastrointestinal diseases and T2DM, we chose the receptors TLR1 (rs10004195) and FCGR2A (rs368433), which are highly correlated with H. pylori IgG antibody seropositivity, as well as six H. pylori antibodies: anti-H. pylori IgG, GroEL, OMP, UREA, CagA, VacA, Catalase, and the three diseases most likely to be induced by H. pylori: chronic gastritis, gastroduodenal ulcer, and gastric cancer as exposure factors, obesity, HbA1c, as well as blood glucose levels were selected as potential mediators, finally, T2DM was selected as an outcome to elucidate the relationship between H. pylori, gastrointestinal diseases and T2DM. For the MR analysis in this work, we employed genetic variants as instrumental factors. Three fundamental presumptions served as the foundation for our MR study's postulated validity: (1) Correlation hypothesis: genetic variation is closely related to exposure; (2) The independence hypothesis: genetic variation is not associated with any confounding factors that may mediate the way from exposure to outcome; (3) Exclusion-restriction hypothesis: Genetic variation can only affect the result through exposure. In order to exclude the influence of heterogeneity, the random effects model (IVW) was used as the main method in this study, and the MR-Egger and weighted median methods were used for verification [15].

### 2. Data sources

Data for the seven antibodies to H. pylori, chronic gastritis, gastroduodenal ulcer, and diabetes and potential mediators were obtained from the GWAS database(https://gwas.mrcieu.ac.uk) [16], and only pooled data from European populations were used to avoid population heterogeneity bias. H. pylori antibody levels of GroEL, OMP, UREA, CagA, VacA and catalase, and blood glucose were obtained from the dataset of the EBI database. Gastroduodenal ulcer, chronic gastritis, gastric cancer, and T2DM were obtained from the Finnish database. Glycated hemoglobin were obtained from the UK Biobank; while the obesity data was collected from GWAS summary datasets. The sample sizes were as Table 1, each GWAS was approved by the appropriate ethics committee.

### 3. Selection of potential genetic variants

Genetic variants can be obtained in two ways, either directly from GWAS summary statistics or through existing reports in the literature. In this study, genes strongly associated with H.

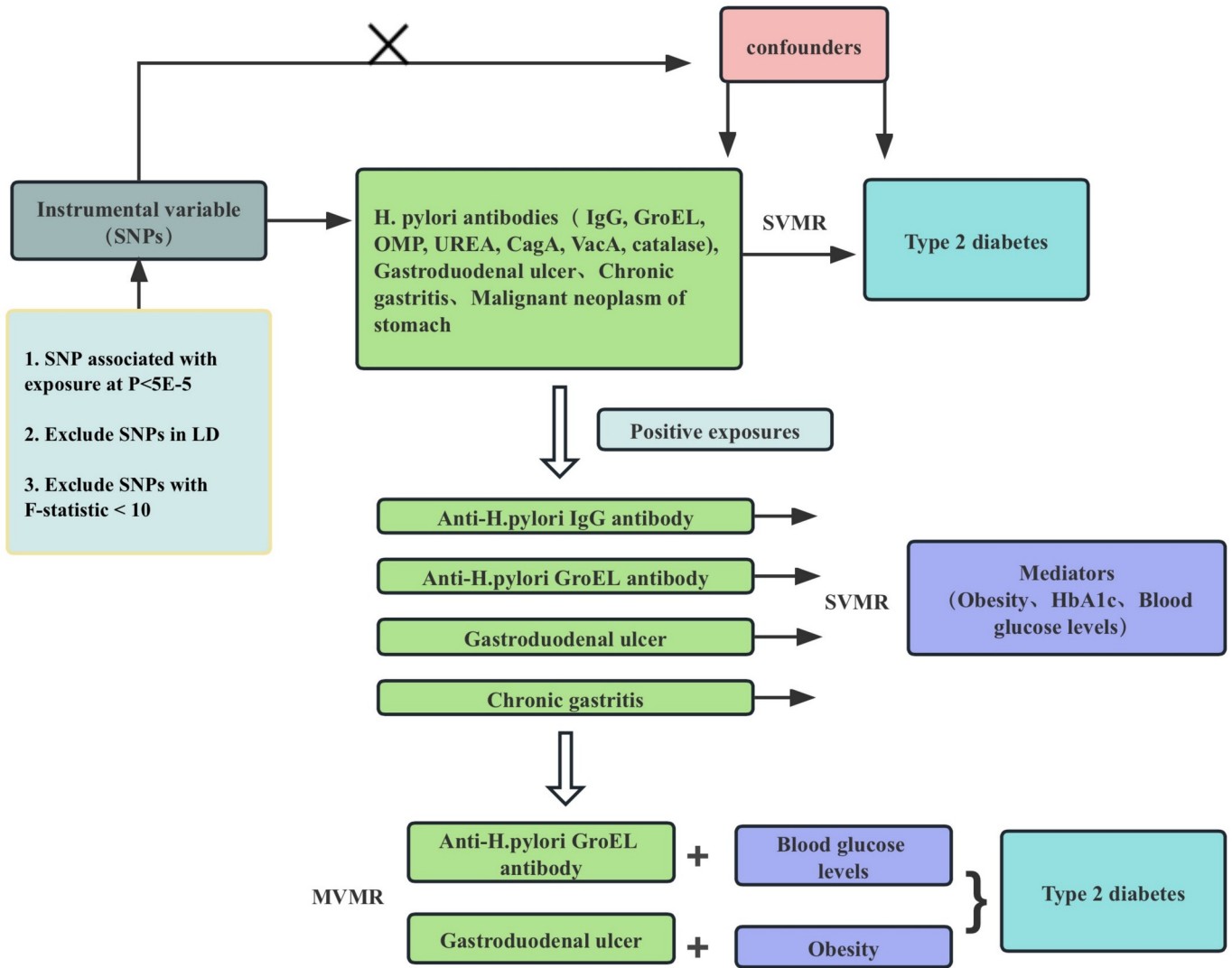

**Fig 1. Flowchart of Mendelian randomization analysis conducted in this study.** SVMR analysis investigates the effect of H. pylori infection and related gastrointestinal diseases on T2DM development. MVMR analysis evaluates the roles of potential factors mediating the association between GroEL-positive H. pylori Infection, gastroduodenal ulcer, chronic gastritis, and T2DM.

pylori infection were selected based on previously available reports and confirming that existing studies have chosen this approach for H. pylori-associated Mendelian studies [17, 18]. The SNPs rs10004195 at TLR1 (4p14) and rs368433 at FCGR 2A gene (1q23.3) have been identified as genetic variants associated with high H. pylori seropositivity [6]. To check the significance of the allelic score as a tool, the F-statistics of both SNPs were calculated to be greater than 10 (Table 2). Other data on H. pylori antibodies, gastrointestinal diseases, T2DM, and potential mediators were obtained from GWAS summary statistics. A variety of tests are necessary to screen for genetic IVs that are eligible and satisfy the MR hypothesis. First, to boost statistical efficacy and get a sufficient number of IVs, we set the p-value threshold for IVs in this MR study to 5E-05 to screen for SNPs that were associated with exposure and strongly associated with exposure. second, independence was set to eliminate linkage disequilibrium (LD: $r^2 =$ 0.001, kb = 10,000, p < 5E-05) and the statistical significance were calculated (F-statistics),

**Table 1. Details of the studies included in the Mendelian randomization analyses.**

| Phenotype | Consortium | Ethnicity | Sample size | GWAS id |
|---|---|---|---|---|
| H. pylori CagA antibody levels | EBI | European | 985 individuals | ebi-a-GCST90006911 |
| H. pylori Catalase antibody levels | EBI | European | 1558 individuals | ebi-a-GCST90006912 |
| H. pylori antibody levels | EBI | European | 2716 individuals | ebi-a-GCST90006913 |
| H. pylori OMP antibody levels | EBI | European | 2640 individuals | ebi-a-GCST90006914 |
| H. pylori UREA antibody levels | EBI | European | 2251 individuals | ebi-a-GCST90006915 |
| H. pylori antibody levels | EBI | European | 1571 individuals | ebi-a-GCST90006916 |
| Gastroduodenal ulcer | FinnGen | European | 4510 cases, 189,695, controls | finn-b-K11_GASTRODUOULC |
| Chronic gastritis | FinnGen | European | 5,213 cases, 189,695 controls | finn-b-K11_CHRONGASTR |
| Malignant neoplasm of stomach | FinnGen | European | 633 cases, 218,159 controls | finn-b-CD2_BENIGN_STOMACH |
| Obesity | GIANT | European | 2896 cases, 47,468 controls | ieu-a-92 |
| HbA1c | UK Biobank | European | 13,586,180 SNPs | ukb-d-30750_raw |
| Blood glucose levels | EBI | European | 400,458 individuals | ebi-a-GCST90025986 |
| Type 2 diabetes | FinnGen | European | 24,133 cases, 183,185 controls | finn-b-E4_DM2NASCOMP |

**Table 2. Instrumental SNPs of IgG-positive H. pylori infection and F statistics.**

| SNP | Beta | SE | EA | NEA | EAF | Pval | F-statistics |
|---|---|---|---|---|---|---|---|
| rs10004195 | 0.3576744 | 0.04048331 | A | T | 0.25 | 1.00E-18 | 78.059 |
| rs368433 | 0.3148107 | 0.05609599 | C | T | 0.16 | 2.00E-08 | 31.495 |

with F-statistics greater than 10 indicates the absence of weak instrumental variable bias [9]. Third, to ensure that the effect alleles are members of the same allele, coordinate the exposure and outcome data sets. These exacting processes can be utilized to screen SNPS, which can then be used as IV for further examination.

## 4. Statistical analysis and data visualization

TwoSampleMR, MR-PRESSO, MVMR and forestploter packages in R software are used for analysis, and inverse variance weighting (IVW) is used as the default method to evaluate the causal estimate [19]. For MR Analysis with more than two IVS, two complementary methods MR-Egger and weighted median are used for verification. A P-value less than 0.05 was considered statistically significant. In order to ensure the reliability of the results, we further carried out sensitivity analysis, that is, heterogeneity and Pleiotropyt test. A P-value less than 0.05 indicates the existence of heterogeneity and pleiotropy. Then, the MR-Egger intercept test is used to evaluate the validity of the hypothesis of instrumental variables. The leave-one-out method is used to analyze whether there are SNPS that drive the results independently, and the MR-PRESSO test is used to evaluate whether there are outlier SNPS. For MVMR, we adjusted for T2DM as an outcome and for glucose levels and obesity, respectively, to explore whether they mediated the causal relationship between anti-H. pylori GroEL and gastroduodenal ulcer, respectively, and T2DM.

## Results

The number of SNPs screened as instrumental variables significantly associated with exposure through LD ranged from 38 to 444 (S1 File)), and the F-statistic for each SNP included in the study was greater than 10 (Table 3).

**Table 3. The number of SNPs and F-statistics of Phenotype.**

| Phenotype | F statistics | nSNP |
|---|---|---|
| H. pylori CagA antibody levels | 19.613 | 90 |
| H. pylori Catalase antibody levels | 18.523 | 80 |
| H. pylori GroEL antibody levels | 18.039 | 68 |
| H. pylori OMP antibody levels | 18.605 | 86 |
| H. pylori UREA antibody levels | 18.691 | 88 |
| H. pylori VacA antibody levels | 18.704 | 93 |
| obesity | 21.959 | 38 |
| HbA1c | 53.192 | 444 |
| blood glucose | 61.164 | 271 |
| Gastroduodenal ulcer | 19.044 | 92 |
| Chronic gastritis | 18.578 | 84 |
| Malignant neoplasm of stomach | 18.693 | 73 |

## Univariate Mendelian randomization

**1. Relationship between potential mediators and T2DM.** The relationship between obesity, HbA1c, blood glucose level and T2DM has been validated in many experiments, and to ensure the feasibility of the Mendelian randomization study, we used the validation by IVW method with genetically predicted obesity (P = 0.023, OR = 1.069, 95%CI = 1.008–1.133), HbA1c (P = 1.263E-76, OR = 1.233, 95% CI = 1.206–1.261), and blood glucose levels (P = 8.21E-42, OR = 4.077, 95% CI = 3.327–4.996) were significantly associated with an increased risk of T2DM. To validate this relationship, we then used the MR Egger method and Weighted median, and the results remained significant (obesity: MR Egger: P = 0.023, OR = 1.254, 95% CI = 1.040–1.512; Weighted median: P = 0.00002, OR = 1.080 95% CI = 1.042–1.119; blood glucose level: MR Egger: P = 6.60E-14, OR = 4.331, 95% CI = 3.013–6.225; Weighted median: P = 1.99E-20, OR = 2.916, 95% CI = 2.325–3.656; HbA1c: MR Egger: P = 5.011E-16, OR = 1.195, 95% CI = 1.147–1.246; Weighted median: P = 3.5E-20, OR = 1.130, 95% CI = 1.101–1.159). In the sensitivity analysis results showed heterogeneity in all three, but no pleiotropy, and none of the egger intercepts significantly deviated from 0. Therefore, obesity, HbA1c, and blood glucose levels can be used as potential mediators for Mendelian randomization analysis (Fig 2).

**2. Effect of H. pylori antibodies on T2DM and its potential mediators.** According to the IVW method, genetically predicted anti-H. pylori igG was substantially linked to a higher risk of T2DM (P = 0.006, b = 0.0945, OR = 1.0995, 95% CI = 1.023–1.176), as well as obesity (P = 0.034, OR = 1.03, 95% CI = 1.002–1.055), but the results with HbA1c (P = 0.105,

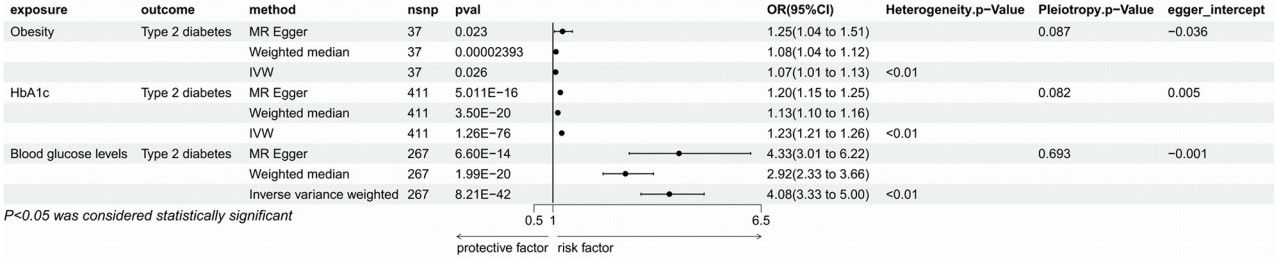

**Fig 2. Mendelian randomization results of the effect of potential mediators (obesity, HbA1c, blood glucose level) on T2DM.**

OR = 0.930, 95% CI = 0.852 to 1.015) were not significant enough for causality, and the test for heterogeneity showed that there was no heterogeneity in these results, which was insufficient for other tests and multivariate analyses due to the limitation of the number of snp. Anti-H. pylori GroEL was significantly in connection with an elevated risk of T2DM by the IVW method (P = 0.028, OR = 1.033, 95% CI = 1.004–1.064), in accordance with the results obtained by the MR Egger (P = 0.04, OR = 1.07, 95% CI = 1.003–1.143) and Weighted median (P = 0.004, OR = 1.06, 95% CI = 1.018–1.096). The sensitivity analysis showed the presence of heterogeneity, and the absence of Pleiotropy (Fig 3), none of the Egger intercepts significantly deviated from 0. The MR-PRESSO results showed the absence of an outlier SNP (S2 File). In addition, anti-H. pylori GroEL was associated with increased blood glucose levels(Fig 4), in agreement with IVW (P = 0.020, OR = 1.011, 95% CI = 1.002–1.021) and Weighted median (P = 0.029, OR = 1.014, 95% CI = 1.001–1.027) methods, In contrast, it was not significant in MR Egger method (P = 0.111, OR = 1.017, 95% CI = 0.997–1.038), Concerning the sensitivity analysis, the results showed no heterogeneity and pleiotropy, none of the egger intercepts significantly deviated from 0, and the MR-PRESSO results also showed the absence of outlying SNP (S2 File), in addition, the leave-one-out method of alignment analysis showed the absence of a single SNP causes the causal consequences of anti-H. pylori GroEL on T2DM and blood glucose levels(Fig 5). However, anti-H. pylori GroEL was not causally associated with obesity and HbA1c, and none of the other antibody- H. pylori were causally associated with T2DM.

**3. Impact of gastrointestinal disorders on T2DM and potential intermediaries.** Genetically predicted gastroduodenal ulcer and chronic gastritis were both in connection with an elevated risk of T2DM, nevertheless, there was no causal relationship between Malignant neoplasm of stomach and T2DM (Fig 3). By using the IVW method, gastroduodenal ulcer was found to be significantly associated with an increased risk of T2DM (P = 0.019, OR = 1.036,

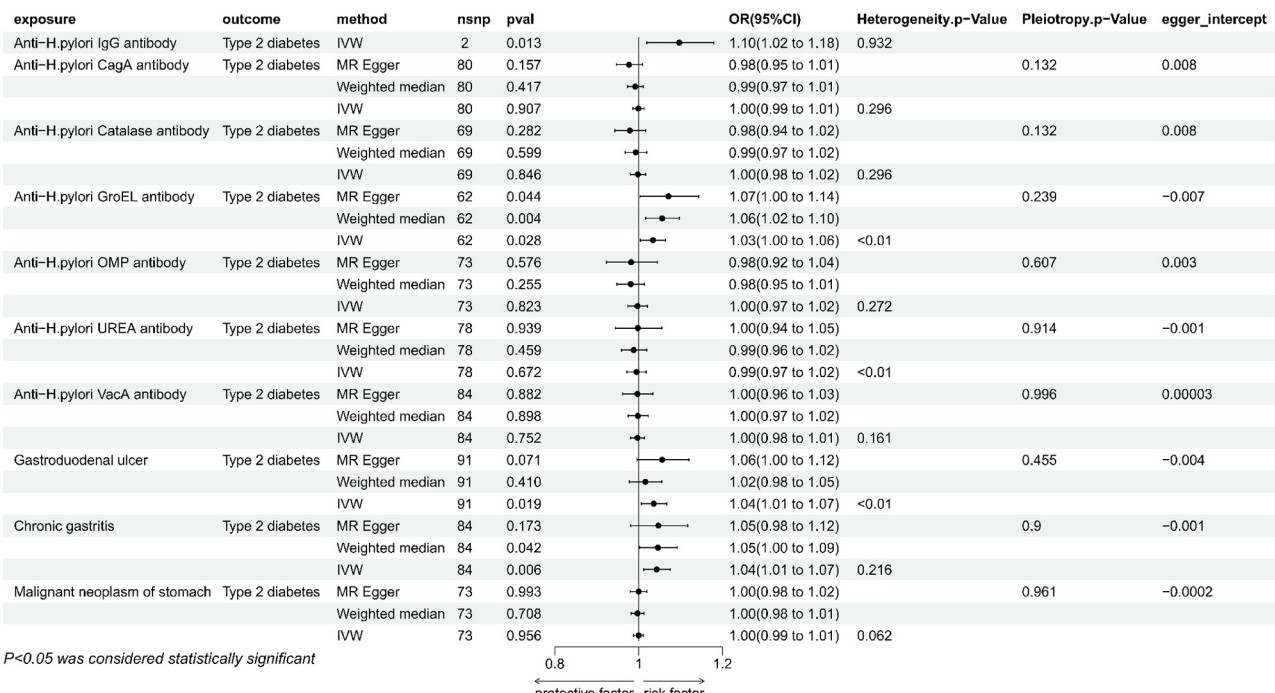

| exposure | outcome | method | nsnp | pval | | OR(95%CI) | Heterogeneity.p-Value | Pleiotropy.p-Value | egger_intercept |
|---|---|---|---|---|---|---|---|---|---|
| Anti-H.pylori IgG antibody | Type 2 diabetes | IVW | 2 | 0.013 | | 1.10(1.02 to 1.18) | 0.932 | | |
| Anti-H.pylori CagA antibody | Type 2 diabetes | MR Egger | 80 | 0.157 | | 0.98(0.95 to 1.01) | | 0.132 | 0.008 |
| | | Weighted median | 80 | 0.417 | | 0.99(0.97 to 1.01) | | | |
| | | IVW | 80 | 0.907 | | 1.00(0.99 to 1.01) | 0.296 | | |
| Anti-H.pylori Catalase antibody | Type 2 diabetes | MR Egger | 69 | 0.282 | | 0.98(0.94 to 1.02) | | 0.132 | 0.008 |
| | | Weighted median | 69 | 0.599 | | 0.99(0.97 to 1.02) | | | |
| | | IVW | 69 | 0.846 | | 1.00(0.98 to 1.02) | 0.296 | | |
| Anti-H.pylori GroEL antibody | Type 2 diabetes | MR Egger | 62 | 0.044 | | 1.07(1.00 to 1.14) | | 0.239 | −0.007 |
| | | Weighted median | 62 | 0.004 | | 1.06(1.02 to 1.10) | | | |
| | | IVW | 62 | 0.028 | | 1.03(1.00 to 1.06) | <0.01 | | |
| Anti-H.pylori OMP antibody | Type 2 diabetes | MR Egger | 73 | 0.576 | | 0.98(0.92 to 1.04) | | 0.607 | 0.003 |
| | | Weighted median | 73 | 0.255 | | 0.98(0.95 to 1.01) | | | |
| | | IVW | 73 | 0.823 | | 1.00(0.97 to 1.02) | 0.272 | | |
| Anti-H.pylori UREA antibody | Type 2 diabetes | MR Egger | 78 | 0.939 | | 1.00(0.94 to 1.05) | | 0.914 | −0.001 |
| | | Weighted median | 78 | 0.459 | | 0.99(0.96 to 1.02) | | | |
| | | IVW | 78 | 0.672 | | 0.99(0.97 to 1.02) | <0.01 | | |
| Anti-H.pylori VacA antibody | Type 2 diabetes | MR Egger | 84 | 0.882 | | 1.00(0.96 to 1.03) | | 0.996 | 0.00003 |
| | | Weighted median | 84 | 0.898 | | 1.00(0.97 to 1.02) | | | |
| | | IVW | 84 | 0.752 | | 1.00(0.98 to 1.01) | 0.161 | | |
| Gastroduodenal ulcer | Type 2 diabetes | MR Egger | 91 | 0.071 | | 1.06(1.00 to 1.12) | | 0.455 | −0.004 |
| | | Weighted median | 91 | 0.410 | | 1.02(0.98 to 1.05) | | | |
| | | IVW | 91 | 0.019 | | 1.04(1.01 to 1.07) | <0.01 | | |
| Chronic gastritis | Type 2 diabetes | MR Egger | 84 | 0.173 | | 1.05(0.98 to 1.12) | | 0.9 | −0.001 |
| | | Weighted median | 84 | 0.042 | | 1.05(1.00 to 1.09) | | | |
| | | IVW | 84 | 0.006 | | 1.04(1.01 to 1.07) | 0.216 | | |
| Malignant neoplasm of stomach | Type 2 diabetes | MR Egger | 73 | 0.993 | | 1.00(0.98 to 1.02) | | 0.961 | −0.0002 |
| | | Weighted median | 73 | 0.708 | | 1.00(0.98 to 1.01) | | | |
| | | IVW | 73 | 0.956 | | 1.00(0.99 to 1.01) | 0.062 | | |

P<0.05 was considered statistically significant

0.8  1  1.2

← protective factor  risk factor →

**Fig 3. Mendelian randomization results of the effect of H. pylori antibodies, gastroduodenal ulcer, chronic gastritis, Malignant neoplasm of stomach on T2DM.**

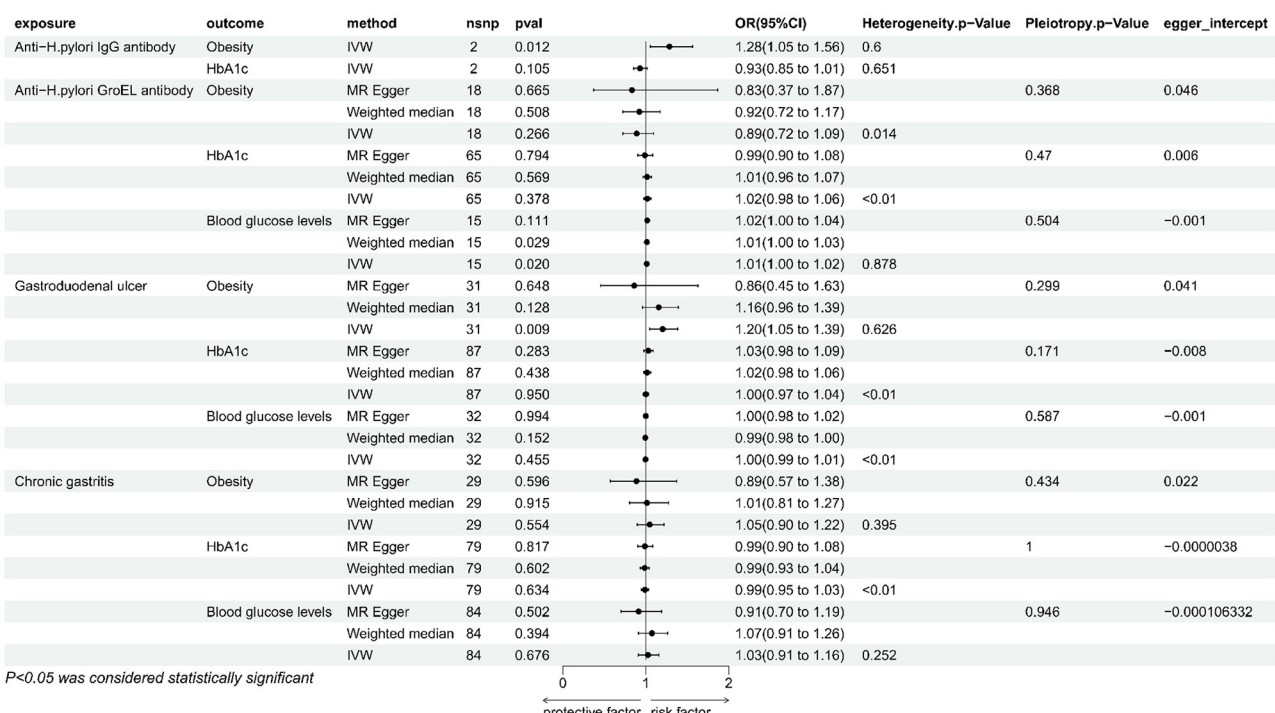

| exposure | outcome | method | nsnp | pval | OR(95%CI) | Heterogeneity.p–Value | Pleiotropy–Value | egger_intercept |
|---|---|---|---|---|---|---|---|---|
| Anti–H.pylori IgG antibody | Obesity | IVW | 2 | 0.012 | 1.28(1.05 to 1.56) | 0.6 | | |
| | HbA1c | IVW | 2 | 0.105 | 0.93(0.85 to 1.01) | 0.651 | | |
| Anti–H.pylori GroEL antibody | Obesity | MR Egger | 18 | 0.665 | 0.83(0.37 to 1.87) | | 0.368 | 0.046 |
| | | Weighted median | 18 | 0.508 | 0.92(0.72 to 1.17) | | | |
| | | IVW | 18 | 0.266 | 0.89(0.72 to 1.09) | 0.014 | | |
| | HbA1c | MR Egger | 65 | 0.794 | 0.99(0.90 to 1.08) | | 0.47 | 0.006 |
| | | Weighted median | 65 | 0.569 | 1.01(0.96 to 1.07) | | | |
| | | IVW | 65 | 0.378 | 1.02(0.98 to 1.06) | <0.01 | | |
| | Blood glucose levels | MR Egger | 15 | 0.111 | 1.02(1.00 to 1.04) | | 0.504 | −0.001 |
| | | Weighted median | 15 | 0.029 | 1.01(1.00 to 1.03) | | | |
| | | IVW | 15 | 0.020 | 1.01(1.00 to 1.02) | 0.878 | | |
| Gastroduodenal ulcer | Obesity | MR Egger | 31 | 0.648 | 0.86(0.45 to 1.63) | | 0.299 | 0.041 |
| | | Weighted median | 31 | 0.128 | 1.16(0.96 to 1.39) | | | |
| | | IVW | 31 | 0.009 | 1.20(1.05 to 1.39) | 0.626 | | |
| | HbA1c | MR Egger | 87 | 0.283 | 1.03(0.98 to 1.09) | | 0.171 | −0.008 |
| | | Weighted median | 87 | 0.438 | 1.02(0.98 to 1.06) | | | |
| | | IVW | 87 | 0.950 | 1.00(0.97 to 1.04) | <0.01 | | |
| | Blood glucose levels | MR Egger | 32 | 0.994 | 1.00(0.98 to 1.02) | | 0.587 | −0.001 |
| | | Weighted median | 32 | 0.152 | 0.99(0.98 to 1.00) | | | |
| | | IVW | 32 | 0.455 | 1.00(0.99 to 1.01) | <0.01 | | |
| Chronic gastritis | Obesity | MR Egger | 29 | 0.596 | 0.89(0.57 to 1.38) | | 0.434 | 0.022 |
| | | Weighted median | 29 | 0.915 | 1.01(0.81 to 1.27) | | | |
| | | IVW | 29 | 0.554 | 1.05(0.90 to 1.22) | 0.395 | | |
| | HbA1c | MR Egger | 79 | 0.817 | 0.99(0.90 to 1.08) | | 1 | −0.0000038 |
| | | Weighted median | 79 | 0.602 | 0.99(0.93 to 1.04) | | | |
| | | IVW | 79 | 0.634 | 0.99(0.95 to 1.03) | <0.01 | | |
| | Blood glucose levels | MR Egger | 84 | 0.502 | 0.91(0.70 to 1.19) | | 0.946 | −0.000106332 |
| | | Weighted median | 84 | 0.394 | 1.07(0.91 to 1.26) | | | |
| | | IVW | 84 | 0.676 | 1.03(0.91 to 1.16) | 0.252 | | |

*P<0.05 was considered statistically significant*

protective factor ← 0    1    2 → risk factor

**Fig 4. Mendelian randomization results of the effect of H. pylori antibodies, gastroduodenal ulcer, chronic gastritis on potential mediators (obesity, HbA1c, blood glucose level).**

95% CI = 1.006–1.068), but not relevant when utilizing the weighted median (P = 0.410, OR = 1.016, 95%CI = 0.978–1.055) or MR-Egger (P = 0.071, OR = 1.056, 95%CI = 0.996–1.120). The assessment of the heterogeneity and pleiotropy, revealed the presence of heterogeneity and absence of pleiotropy. The egger intercept did not significantly deviate from 0, the leave-one-out permutation analyses revealed the lack of a single SNP with a causal effect on T2DM, the MR-PRESSO results also revealed the absence of outlier SNPs (S2 File)). Furthermore, the gastroduodenal ulcer was also linked to an increased risk of obesity as determined by the IVW method (P = 0.009, OR = 1.204, 95% CI = 1.047 to 1.385), but was not significant in the other two methods, there was no heterogeneity and pleiotropy (Fig 4), and the leave-one-out method showed the absence of individual SNPs driving the causal effect on obesity (Fig 5). According to the results, the gastroduodenal ulcer was not causally associated with HbA1c and blood glucose levels. Regarding the chronic gastritis lesions, the results showed a significant link between chronic gastritis and increased risk of T2DM (P = 0.005, OR = 1.042, 95% CI = 1.012–1.074), and the result agreed with weighted median approach (P = 0.042, OR = 1.046, 95% CI = 1.002–1.092). While in MR Egger the result was not significant, sensitivity analysis showed no heterogeneity or pleiotropy, and the Egger intercept was almost close to 0, the leave-one-out method also showed that there was no snp that individually drives a causal effect on T2DM (Fig 5), MR-PRESSO results showed that there was no standing group of snp (S2 File). No causal relationship between chronic gastritis and any of the three candidate mediators was revealed (Fig 4).

## Multivariate Mendelian randomization

**1. Mediating relationship between anti-H. pylori GroEL and blood glucose levels with T2DM.** MVMR was carried out to determine whether blood glucose levels were a mediating

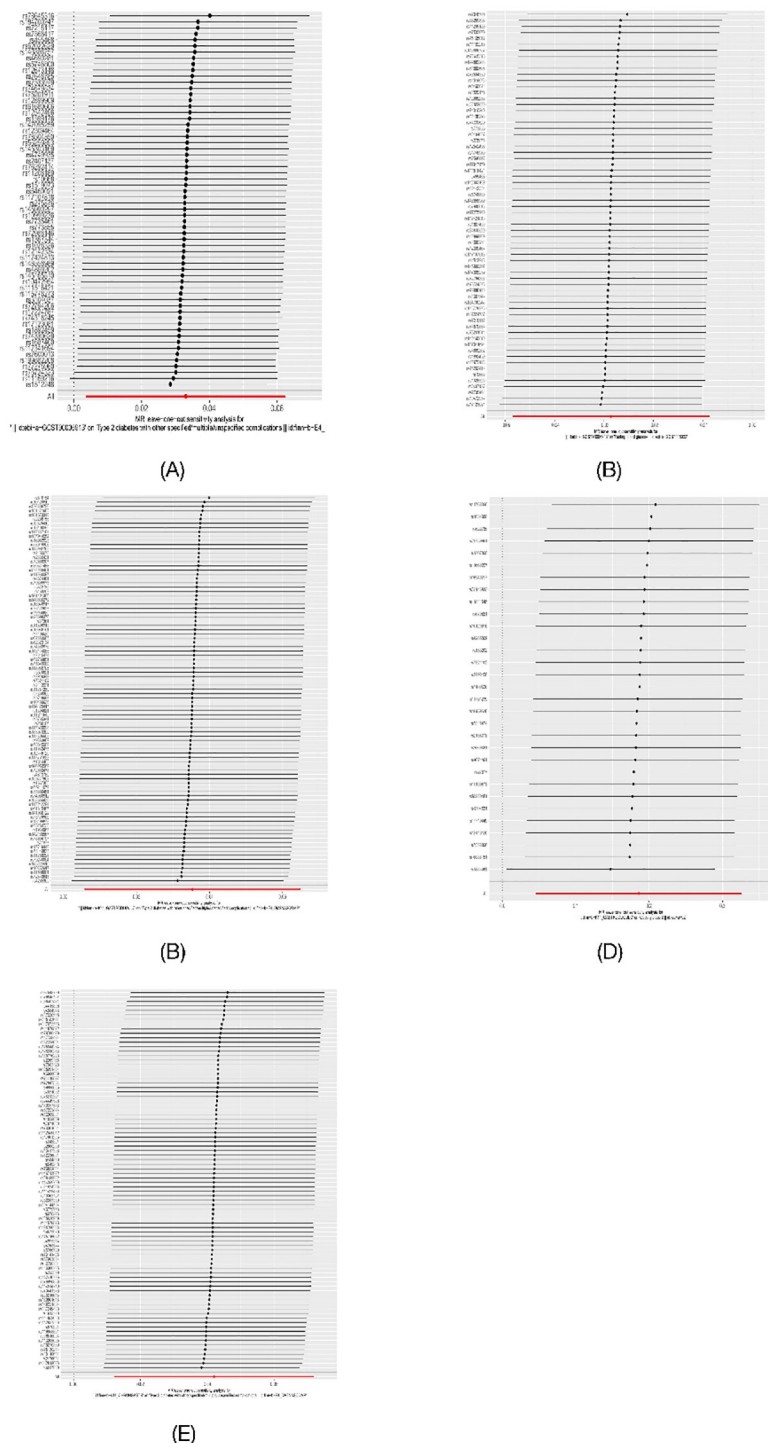

**Fig 5.** (A, B) Mendelian randomization result of the leave-one-out method of anti-H.pylori GroEL on T2DM and blood glucose levels. (C, D) Mendelian randomization result of the leave-one-out method of gastroduodenal ulcer on T2DM and obesity. (E) Mendelian randomization results of the leave-one-out method of chronic gastritis on T2DM.

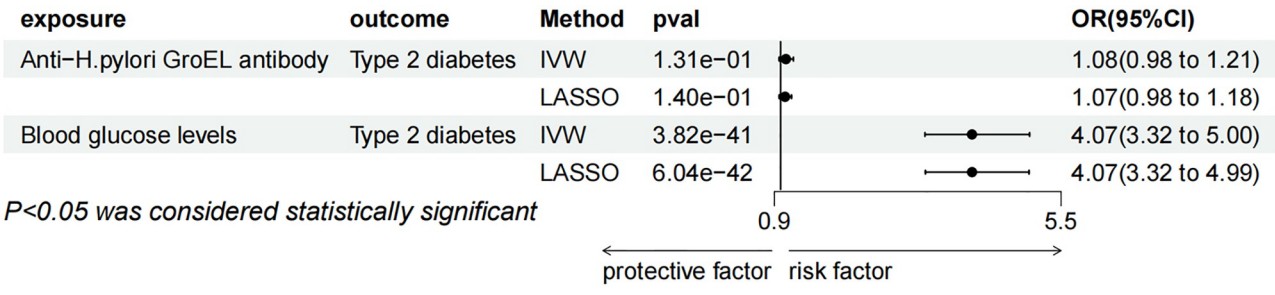

**Fig 6. Multivariable MR result of causal relationships of anti-H.pylori GroEL and blood glucose levels on T2DM.**

factor in the association between GroEL antibody-positive H. pylori and T2DM. When blood glucose levels were adjusted, the connection between T2DM and anti-H. pylori GroEL vanished (P = 0.131, OR = 1.085, 95% CI = 0.976–1.205) in IVW method, but the link between blood glucose levels and the disease was still persisted (P = 3.82E-41, OR = 4.071, 95% CI = 3.317–4.996). The results of MVMR using Lasso methods showed consistent results with the IVW method (Fig 6).

**2. Mediating relationship between Gastroduodenal ulcer and obesity with T2DM.** The relationship between T2DM and obesity-mediated gastroduodenal ulcer was examined using MVMR. The results revealed that while the relationship between gastroduodenal ulcer and T2DM vanished after combining the instrumental variables of obesity and gastroduodenal ulcer, the association between obesity and T2DM remained substantial. (P = 0.541, OR = 1.033, 95% CI = 0.931–1.146). Similarly, the MVMR results of Lasso were consistent with IVW method (Fig 7).

## Discussion

Although evidence indicates that H. pylori infection is linked to a higher likelihood of T2DM, the exact cause and underlying mechanisms remain unknown. We investigated the causal relationships among seven different types of H. pylori antibodies, gastroduodenal ulcer, chronic gastritis, gastric cancer, and T2DM by using univariate Mendelian randomization, and we investigated whether these causal relationships were mediated using multivariate Mendelian randomization. The results showed that the relationship among anti-H. pylori IgG, anti-H. pylori Groel, gastroduodenal ulcer, chronic gastritis and type 2 diabetes is causally related. In addition, fasting glucose and obesity, respectively, act as mediators in the relationships between anti-H. pylori Groel, gastroduodenal ulcer, and T2DM.

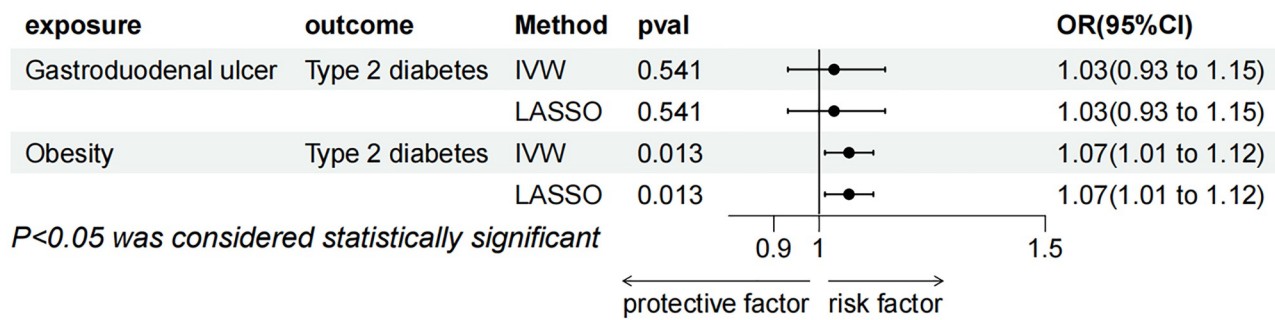

**Fig 7. Multivariable MR result of causal relationships of gastroduodenal ulcer and obesity on T2DM.**

Antibodies anti-H. pylori are found in the infected patients. It is reported, that a higher levels of H. pylori antibodies anti-H. pylori is linked to T2DM [20]. An observational study found that more than 75% of diabetic patients had elevated H. pylori antibodies and from gastric problems [10]. SNP rs 10004195 of the Toll-like receptor 1 (TLR 1) gene at 4p 14 and SNP rs368433 of the FCGR 2A gene at 1Q 23.3 have been identified as genetic variants with the strongest association for H. pylori sero-positive rates [6]. TLR1's A allele has been linked to an increase in H. pylori infections, and FCGR polymorphisms have been linked to a number of persistent bacterial infections [18, 21]. By analyzing the original literature, it was discovered that these two receptors were taken from H. pylori serum igG antibodies. These two receptors were studied by Mendelian randomization in prior relevant studies on behalf of H. pylori as an exposure factor [17, 22]. Mendelian study of these two receptors with T2DM and potential mediators revealed a strong association between them and an elevated risk of both T2DM and obesity. However, due to the limited number of SNPs, it was not sufficient to carry out the next step of the multivariate Mendelian randomization study; thus, we reasonably hypothesized by combining previous reports in the literature and our findings that obesity may be a mediating factor in the increase of T2DM by anti-H. pylori igG, and our speculation is consistent with a study on the correlation of H. pylori and obesity with T2DM, which indicated that the prevalence of H. pylori infection is increased in obese patients with T2DM, and obese patients with T2DM had a greater rate of H. pylori infection [9].

GroEL is a chaperone protein that is necessary for the proper folding of several proteins in bacteria [23]. Previous research has revealed that GroEL is a novel H. pylori virulence factor that is extensively expressed in most H. pylori strains and may act as a possible indicator for high-risk H. pylori infections. Antibodies against GroEL are linked to H. pylori infections and are capable of sticking to gastric epithelial cells and triggering inflammatory responses [24], which is an independent predictor of gastric lesion progression [25]. Furthermore, GroEL is cross-reactive in humans and H. pylori, and this cross-reactive antibody may contribute to the inflammatory response, raising the risk of T2DM [26]. Our MR study suggest that GroEL antibodies not only increase the risk of T2DM, but also raise blood glucose levels, which is similar with the findings of Ningning You et al. on H. pylori and blood glucose [27], who demonstrated that H. pylori infection is a distinct risk factor for elevated blood glucose levels in nondiabetic individuals, and that persistent H. pylori infection leads to elevated fasting plasma glucose and TG/HDL levels, which give rise to the development of T2DM. This could be because H. pylori infection increases the consumption of a high-fat diet, which alters the diversity of key intestinal bacteria, causing an imbalance in intestinal homeostasis, increasing lipid metabolism, and ultimately leading to an imbalance in glucose metabolism [28], and previous research has shown that intestinal microbiology plays a significant role in the hyperglycemia brought on by H. pylori [29]. Therefore, it is imperative to get eradicated of H. pylori infections as soon as they are found in order to prevent T2DM and blood glucose abnormalities from developing as a result of chronic infections.

The role of H. pylori infection in peptic ulcer, gastric cancer, gastritis, and other gastrointestinal disorders is becoming more and more clearly recognized with the passage of time [30]. In an observational study, as many as 75% of diabetic patients responded with significant gastrointestinal symptoms [31, 32], presenting with dysphagia, reflux, constipation, stomach discomfort, nausea, vomiting, and diarrhea [8]. It has been reported that the release of gastric-related hormones may be influenced by H. pylori-induced gastritis, and among these hormones, leptin, hunger hormone, gastrin, and growth inhibitor all affect the susceptibility to diabetes and promote obesity and diabetes [33, 34]. The following are some potential mechanisms at play: High levels of leptin may hamper function of pancreatic islet, inhibit insulin production in response to glucose and cause human pancreatic B-cell death via activating c-JNK [35]; gastrin increases food-related and glucose-stimulated insulin release [36], growth inhibitor controls pancreatic insulin

secretion and prevents insulin release, while starvation hormone lowers energy expenditure and encourages weight gain [37]. In our study, we carried out a Mendelian randomization study of the three diseases most likely to be induced by H. pylori with T2DM, and the results showed that gastric cancer was not causally related to T2DM, while gastroduodenal ulcer and chronic gastritis were associated with the development of T2DM, and in order to further explore the mechanism of the occurrence of this causality, we compared the two diseases with the T2DM development, respectively. Mendelian analysis of obesity, HbA1c and blood glucose levels, respectively, which showed that gastroduodenal ulcer was a risk factor for increased risk of obesity, so we hypothesized that obesity might be a mediator between gastroduodenal ulcer and T2DM, and then we carried out the validation, and the result was in line with our speculation that gastroduodenal ulcer mediated T2DM through obesity, which was in line with the results of previous results of a Mendelian study on the association between peptic ulcer and obesity [38], therefore, obese patients with gastroduodenal ulcer should be carefully managed to prevent T2DM.

In conclusion, among the exposures adopted in this study, anti-H. pylori IgG, GroEL, gastroduodenal ulcer and chronic gastritis were causally associated with T2DM, which suggests a close relevance between gastrointestinal disorders and the condition. Patients with H. pylori should pay close attention to T2DM-related indicators to achieve early detection and prompt treatment. Additionally, this study and other article reports highlight the crucial relationship between obesity and T2DM, thus people with H. pylori infection must pay attention to weight control to stop the condition from progressing.

The strengths of this study are the selection of data from larger studies, the full inclusion of indicators of H. pylori infection and associated gastrointestinal disorders as exposures, the first exploration of the association between H. pylori and T2DM by Mendelian randomization, and the use of multivariate MR analyses to explore mediating pathways in order to identify a number of possible mechanisms, filling a gap in the randomized controlled trials. There are also some limitations of this study. First, to obtain sufficient instrumental variables, the P-value threshold chosen for IV was 5E-05, which may introduce a weak instrumental bias to the overall estimates. Second, there is a small dataset on H. pylori infection in the GWAS database, and although we used snps and antibodies strongly associated with it to do the study and get positive results, there is still a need for more comprehensive data to line up a broader study on H. pylori. Third, in order to avoid the bias of population heterogeneity, we only based the GWAS summary statistics on the population of European descent, and the applicability of these results to other ethnic groups needs to be further explored.

## Conclusion

Our research discovered that H. pylori IgG antibody, H. pylori GroEL antibody, gastroduodenal ulcer, and chronic gastritis are all linked to type 2 diabetes. Additionally, blood glucose levels and obesity serve as intermediaries between H. pylori GroEL antibody, gastroduodenal ulcerand type 2 diabetes, respectively. This study indicates that H. pylori eradication therapy can decrease the risk of developing type 2 diabetes. Furthermore, controlling blood sugar and weight can help reduce the risk of developing type 2 diabetes in individuals with H. pylori infection and gastroduodenal ulcer.

## Supporting information

**S1 File. SNPS information for tool variables.** SNPS for Mendelian randomization analysis after linkage imbalance and adjustment of p-values for all instrumental variables.
(ZIP)

**S2 File. Results of MRPRESSO by R software.** The original MRPRESSO result obtained by R software.
(DOCX)

## Acknowledgments

We gratefully acknowledge the authors and participants of all GWASs from which we used summary statistics data.

## Author Contributions

**Conceptualization:** Mei Sun, Zhe Zhang.

**Data curation:** Mei Sun, Juewei Zhang.

**Formal analysis:** Zhe Zhang.

**Funding acquisition:** Shouyu Wang.

**Investigation:** Jingjing Zhang, Juewei Zhang, Zhuqiang Jia, Lin Zhao.

**Methodology:** Xiaohong Sun.

**Project administration:** Lin Zhao, Xiaohong Sun, Junwei Zong.

**Resources:** Ying Zhu.

**Software:** Zhuqiang Jia, Xin Han.

**Supervision:** Junwei Zong, Ying Zhu.

**Validation:** Mei Sun.

**Visualization:** Jingjing Zhang.

**Writing – original draft:** Zhe Zhang.

**Writing – review & editing:** Shouyu Wang.

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
