## [Decision Letter · Decision Letter 0]

31 Jan 2024

PONE-D-23-39158Causal relationships of Helicobacter pylori and related gastrointestinal diseases on Type 2 diabetes: Univariable and Multivariable Mendelian randomizationPLOS ONE

Dear Dr. wang,

Thank you for submitting your manuscript to PLOS ONE. After careful consideration, we feel that it has merit but does not fully meet PLOS ONE’s publication criteria as it currently stands. Therefore, we invite you to submit a revised version of the manuscript that addresses the points raised during the review process.

We look forward to receiving your revised manuscript.

Kind regards,

Yasin Sahin

Academic Editor

PLOS ONE

Journal Requirements:

This study was supported by the National Natural Science Foundation of China (82074426, 82104864, 82204822), Natural Science Foundation of Liaoning Province (2021-BS-215, 2022-MS-25, 2023-MS-13), Liaoning Revitalization Talents Program (XLYC1802014), Liaoning Key Research and Development Planning Project (2017226015), Basic Research Projects of Liaoning Provincial Department of Education (LJKMZ20221286), Naural Science Foundation of Tibet Autonomous Region and Regional Science(XZ202301ZR0030G, XZ2023ZR-ZY82(Z)) and Technology Project of Naqu City.

Additional Editor Comments:

Thank you for the study. I invite you to resubmit your manuscript after addressing two reviewers’ comments. When resubmitting your manuscript, please carefully consider all issues mentioned in the reviewers' comments, outline every change made point by point, and provide suitable rebuttals for any comments not addressed.

Reviewers' comments:

Reviewer's Responses to Questions

**Comments to the Author**

1. Is the manuscript technically sound, and do the data support the conclusions?

Reviewer #1: Partly

Reviewer #2: Partly

2. Has the statistical analysis been performed appropriately and rigorously? 

Reviewer #1: I Don't Know

Reviewer #2: No

3. Have the authors made all data underlying the findings in their manuscript fully available?

Reviewer #1: Yes

Reviewer #2: Yes

4. Is the manuscript presented in an intelligible fashion and written in standard English?

Reviewer #1: No

Reviewer #2: No

5. Review Comments to the Author

Reviewer #1: Authors have done a commendable job in preparing this manuscript. I am unable to comment on statical analysis. Ref. no 5 should be looked at and corrected. The related statement saying " Sweden " should be corrected. All other ref. should be reviewed and make sure the statements are correct.

The manuscript should be corrected by an English-speaking writer.

Reviewer #2: The study design is interesting by using several statistical methods to assess the relationship between H.pylori infection and the pathologies associated, and other mediators with the risk of developing type 2 diabetes. However, the writing of the article is not good, difficult to read and understand, requirs a revision, use of simple and scientific words. I've attached the article with remarks marked in red and green.

*Study concept: good

*Statistical analyses: good

*Writting: difficult to read and understand, grammatical errors, revision is required (article not ready for publication).

6. PLOS authors have the option to publish the peer review history of their article (what does this mean?). If published, this will include your full peer review and any attached files.

Reviewer #1: No

Reviewer #2: **Yes: **Dr. Ghizlane Bounder

---

## [Author Response · Author response to Decision Letter 0]

1 Mar 2024

Dear Editor and Reviewers,

First and foremost, we would like to express our heartfelt gratitude for your recent correspondence. We are sincerely thankful for the constructive feedback provided by the reviewers regarding our manuscript. Your insights have been immensely valuable to us as we continue to refine and improve our work. The insights offered have been immensely valuable and have greatly contributed to the enhancement of our paper. In light of the reviewers' suggestions, we have undertaken extensive revisions aimed at strengthening the persuasiveness of our work. The changes made to the manuscript are reflected in the Revised manuscript (marked-up copy), and we have outlined our responses to the comments from our two kind and approachable reviewers below:

Journal Requirements:

1.When submitting your revision, we need you to address these additional requirements.

We ensure that the format of the revised article meets the requirements of the journal.

Following the resubmission, the 'Funding Information' section has been duly provided and corrected.

We have provided a revised statement in our cover letter stating all sources of funding or support. 

Additional Editor Comments:

Thank you for the study. I invite you to resubmit your manuscript after addressing two reviewers’ comments. When resubmitting your manuscript, please carefully consider all issues mentioned in the reviewers' comments, outline every change made point by point, and provide suitable rebuttals for any comments not addressed.

We are profoundly grateful for your meticulous review of our manuscript. Your concerns have been duly noted, and we are appreciative of the insightful suggestions that have greatly contributed to improving our work. In response to your feedback, we have undertaken comprehensive revisions to the earlier draft, ensuring that the issues raised have been thoroughly addressed.

COMMENTS TO THE AUTHOR:

1.Is the manuscript technically sound, and do the data support the conclusions?

Reviewer #1: Partly

Reviewer #2: Partly

We are grateful for your careful review and evaluation of the content of our manuscript. We highly value the concerns you have raised regarding the technical soundness of our paper and the support of data for our conclusions.

Our research is methodologically sound, and the data derived robustly support the conclusions drawn. Our study employed the Mendelian randomization approach, a causal inference method based on genetic variants. Its technical reliability hinges on three core assumptions: the relevance assumption ensures a strong correlation between the chosen SNPs and the exposure factor; the independence assumption requires that the SNPs are independent of confounding factors; and the exclusion restriction assumption suggests that the SNPs affect the outcome solely through the exposure. These assumptions collectively underpin the theoretical foundation of Mendelian randomization, enabling researchers to leverage the natural random assortment of genes to investigate causal relationships between specific biological factors and diseases.

In terms of the extent to which the data bolster the conclusions, Mendelian randomization studies utilize genetic data as instrumental variables, akin to the randomization process in randomized controlled trials, thereby aiding scientists in exploring the causal links between an exposure and an outcome. This methodology allows researchers to employ large, publicly available genome-wide association datasets for causal inference, offering fresh avenues for research in fields such as neurology. A key advantage of the Mendelian randomization method lies in its effect estimates being unaffected by confounding factors and reverse causation, thus providing clearer evidence to substantiate particular conclusions. In summary, the Mendelian randomization method ensures the scientific integrity and credibility of its analysis through rigorous hypothesis testing and analytical procedures.

We would like to express our gratitude once more for your invaluable feedback, which is essential for enhancing the quality of our manuscript.

2.Has the statistical analysis been performed appropriately and rigorously?

Reviewer #1: I Don't Know

Reviewer #2: No

We are deeply appreciative of the review process and the insights provided by the reviewers. Recognizing the importance of statistical rigor in our manuscript, we take the feedback seriously, especially given that one reviewer indicated uncertainty and the other noted concerns with our statistical analysis. Our research has incorporated appropriate and rigorous statistical analyses. Mendelian randomization (MR) relies on a series of assumptions, including the relevance of genetic variants to risk factors, the independence of genetic variants from confounding factors, and the exclusion restriction that genetic variants affect outcomes only through the risk factors. The validation and analysis of these assumptions are closely related to statistics. In our study, we have conducted: (1) Relevance assessment: We examined the strength of association between genetic variants and risk factors through MR methods, using inverse-variance weighting (IVW) to estimate the effect of exposure on outcomes by weighted averaging the effect sizes of each genetic variant, with weights typically based on the inverse of the standard error of each variant's effect size (i.e., inverse variance), and regression analysis with the MR-Egger method to adjust for directional pleiotropy. (2) Independence and exclusion restriction tests: In our analysis, we employed sensitivity analyses such as heterogeneity and pleiotropy tests to verify the reliability of our results. (3) Causal inference analysis: At the heart of MR studies is the use of genetic data as an instrumental variable to explore the causal relationship between exposure and outcomes. This involves statistical methods such as linear regression and logistic regression to estimate the impact of genetic variants on disease risk. Therefore, in MR studies, the results of statistical analysis are correctly interpreted and presented, ensuring the scientific integrity and reliability of the research. The application of these methods enables MR studies to more precisely estimate the causal relationships between exposure factors and diseases, helping to overcome issues of confounding and reverse causation inherent in traditional observational studies. Through these advanced statistical techniques, MR studies provide robust evidence to support public health policy formulation and clinical practice.

Overall, statistical analysis plays a crucial role in the Mendelian randomization method; it not only helps to validate key assumptions but also serves to estimate and explain the influence of genetic variants on disease risk. Through these analyses, researchers can gain a better understanding of the causal relationships between biological factors and diseases.

Thank you once again for your constructive criticism, which plays a crucial role in enhancing the quality of our research.

3.Have the authors made all data underlying the findings in their manuscript fully available?

Reviewer #1: Yes

Reviewer #2: Yes

We are deeply grateful for your feedback and affirmation. Indeed, we have ensured that all pertinent data underlying the findings in our manuscript are fully accessible to the public. The availability of these data is critical for the replication of our results and for fostering further research endeavors.

Thank you once again for your support and engagement with our work.

 4. Is the manuscript presented in an intelligible fashion and written in standard English?

Reviewer #1: No

Reviewer #2: No

We are truly grateful for your careful review and constructive feedback regarding our manuscript. We acknowledge that, as pointed out, there is room for improvement in terms of clarity and adherence to standard English. Rest assured, we will undertake a thorough revision to enhance the clarity and readability of our document, ensuring it aligns with the norms of standard English. We are extremely grateful to Reviewer #2 for taking the valuable time to polish the language of this article. We have made our best efforts to improve the manuscript, refining the text throughout based on the reviewer's invaluable suggestions. These changes do not affect the content and framework of the paper; the modifications made after polishing are listed in the Revised manuscript (marked-up copy). Here, we would like to extend our heartfelt thanks once again to the reviewer for their enthusiastic work, and we hope that these revisions will be approved.

5. Review Comments to the Author

Reviewer #1: Authors have done a commendable job in preparing this manuscript. I am unable to comment on statical analysis. Ref. no 5 should be looked at and corrected. The related statement saying " Sweden " should be corrected. All other ref. should be reviewed and make sure the statements are correct. The manuscript should be corrected by an English-speaking writer.

We extend our heartfelt gratitude for your thorough review and the evaluation of our manuscript. We sincerely appreciate the invaluable suggestions and feedback you have provided throughout the review process. Thank you very much for your valuable feedback. Taking into account your suggestions, we have polished the language of the article in conjunction with the corrections proposed by a native English-speaker and have uploaded the revised paper. Thanks again for your advice, which avoids mistakes and makes the article more perfect. We wish you success in all your endeavors!

Reviewer #2: The study design is interesting by using several statistical methods to assess the relationship between H.pylori infection and the pathologies associated, and other mediators with the risk of developing type 2 diabetes. However, the writing of the article is not good, difficult to read and understand, requirs a revision, use of simple and scientific words. I've attached the article with remarks marked in red and green.

We are deeply grateful for your meticulous review and appraisal of our manuscript. We are particularly honored by your recognition of the intriguing study design, which is greatly appreciated. Moreover, we extend our heartfelt thanks for the invaluable suggestions and edits you have contributed towards refining the article. We are extremely grateful for your time in reviewing this article and especially for your help in polishing its language. We have revised the article according to your suggestions, and the updated version has been uploaded. Thank you once again for your valuable input and for the language improvements you've made. Wish you the best!

6. PLOS authors have the option to publish the peer review history of their article (what does this mean?). If published, this will include your full peer review and any attached files.

Yes, we would like to publish the peer review history.

In closing, we wish to express our heartfelt thanks once more for your diligent work. The professional opinions and guidance you have provided are not only instrumental in improving the quality of our manuscript but also serve as invaluable direction for our research trajectory. We extend our deepest respect for your dedication and contributions.

We eagerly look forward to further guidance and feedback on our revised manuscript.

With warm regards,

Shouyu Wang

---

## [Editor Report · Decision Letter 1]

6 Mar 2024

Causal relationships of Helicobacter pylori and related gastrointestinal diseases on Type 2 diabetes: Univariable and Multivariable Mendelian randomization

PONE-D-23-39158R1

Dear Dr. Shouyu wang,

We’re pleased to inform you that your manuscript has been judged scientifically suitable for publication and will be formally accepted for publication once it meets all outstanding technical requirements.

Kind regards,

Yasin Sahin

Academic Editor

PLOS ONE

Additional Editor Comments (optional):

Thank you for the study. The authors did an appropriate point-by-point response to the reviewers.
---

## [Editor Report · Acceptance letter]

3 Apr 2024

PONE-D-23-39158R1 

PLOS ONE

Dear Dr. wang, 

I'm pleased to inform you that your manuscript has been deemed suitable for publication in PLOS ONE. Congratulations! Your manuscript is now being handed over to our production team.

Kind regards, 

on behalf of

Dr. Yasin Sahin 

Academic Editor

PLOS ONE